# Different Patterns of Left Ventricular Hypertrophy in Metabolically Healthy and Insulin-Resistant Obese Subjects

**DOI:** 10.3390/nu12020412

**Published:** 2020-02-05

**Authors:** Angela Sciacqua, Antonio Cimellaro, Luana Mancuso, Sofia Miceli, Velia Cassano, Maria Perticone, Teresa V. Fiorentino, Francesco Andreozzi, Elena Succurro, Giorgio Sesti, Francesco Perticone

**Affiliations:** 1Department of Medical and Surgical Sciences, University Magna Græcia of Catanzaro, 88100 Catanzaro, Italy; luana.mancuso@hotmail.it (L.M.); sofy.miceli@libero.it (S.M.); velia.cassano@libero.it (V.C.); vanessa.fiorentino@unicz.it (T.V.F.); andreozzif@unicz.it (F.A.); succurro@unicz.it (E.S.); perticone@unicz.it (F.P.); 2Internal Medicine Unit, Pugliese-Ciaccio Hospital, 88100 Catanzaro, Italy; antocime@hotmail.it; 3Department of Experimental and Clinical Medicine, University Magna Græcia of Catanzaro, 88100 Catanzaro, Italy; mariaperticone@hotmail.com; 4Department of Clinical and Molecular Medicine, University of Rome-Sapienza, 00161 Rome, Italy; giorgio.sesti@uniroma1.it

**Keywords:** obesity, metabolically healthy obesity, insulin-resistance, left ventricular hypertrophy, cardiac remodeling

## Abstract

Obese subjects showed different cardiovascular risk depending by different insulin sensitivity status. We investigated the difference in left ventricular mass and geometry between metabolically healthy (MHO) and unhealthy (MUHO) obese subjects. From a cohort of 876 obese subjects (48.3 ± 14.1 years) without cardio-metabolic disease and stratified according to increasing values of Matsuda index after 75 g oral glucose tolerance test, we defined MHO (*n* = 292) those in the upper tertile and MUHO (*n* = 292) those in the lower tertile. All participants underwent echocardiographic measurements. Left ventricular mass was calculated by Devereux equation and normalized by height^2,7^ and left ventricular hypertrophy (LVH) was defined by values >44 g/m^2.7^ for females and >48 g/m^2.7^ for males. Left ventricular geometric pattern was defined as concentric or eccentric if relative wall thickness was higher or lower than 0.42, respectively. MHO developed more commonly a concentric remodeling (19.9 vs. 9.9%; *p* = 0.001) and had a reduced risk for LVH (OR 0.46; *p* < 0.0001) than MUHO, in which the eccentric type was more prevalent (40.4 vs. 5.1%; *p* < 0.0001). We demonstrated that obese subjects—matched for age, gender and BMI—have different left ventricular mass and geometry due to different insulin sensitivity status, suggesting that diverse metabolic phenotypes lead to alternative myocardial adaptation.

## 1. Introduction

Obesity represents a crucial problem for public health due to reach epidemic proportion worldwide and to strongly associate with an increased risk for type 2 diabetes mellitus and cardiovascular (CV) disease, both conditions able to worse clinical outcome [1]. However, there is increasing evidence that sub-phenotypes of obesity exist, thus questioning the apparent linear relationship between increased body mass index (BMI) and cardio-metabolic risk [2,3,4]. In fact, approximately from 10% to 25% of obese subjects are free of any metabolic abnormalities, are relatively insulin sensitive and have a rather favorable CV risk profile, thus they exhibit metabolically healthy obesity (MHO) [3,4,5]. This wide range may be justified by the fact that, despite the enhanced interest in MHO, there are not unique criteria to define MHO, thus the implications of MHO phenotype on CV risk is still not clear.

It is known that left ventricular hypertrophy (LVH) represents an independent risk factor for CV morbidity and mortality in different clinical settings [6,7]. However, the increase of left ventricular mass (LVM) is not only a consequence of adaptive cardiac remodeling to hemodynamic overload, but it recognizes a complex and multifactorial pathogenesis. In particular, relevant studies have demonstrated that metabolic factors have an important role in the cardiac remodeling and LVM increase [8,9,10,11,12,13,14]. Several evidences have demonstrated that obesity is an independent risk factor for LVH [15,16]. In fact, independently of arterial hypertension, increased adiposity may promote structural and functional changes in the myocardium through hemodynamic and non-hemodynamic factors [17]. According with this, LVH and left ventricular diastolic dysfunction are present in young obese before sustained hypertension development [18]. In consideration of the different obesity phenotypes, it is possible to speculate that it is not obesity itself, but the obesity-related metabolic abnormalities that affect LVM and LVH development with possible consequence on clinical outcome.

Currently, there are few data about the association between MHO and LVH and this issue remains to be elucidated. Thus, the aim of the present study is to evaluate whether various obesity phenotypes may affect, in a different way, left ventricular remodeling.

## 2. Materials and Methods

### 2.1. Study Population

From the CATAnzaro MEtabolic RIsk factors Study (CATAMERIS), a prevention campaign for cardio-metabolic risk factors performed at University “Magna Graecia” of Catanzaro [19], an initial cohort of 876 Caucasian obese subjects was considered. All participants, aged 48.3 ± 14.1 years (380 males, 496 females), showed a BMI ≥ 30 kg/m^2^, without history of CV and metabolic disease, and after a 12-h overnight fast underwent a 75 g oral glucose tolerance test (OGTT) with 0, 30-, 60-, 90- and 120-min sampling for plasma insulin and glucose. Then, the cohort was stratified into tertiles according to increasing values of Matsuda index, a previously validated insulin sensitivity parameter derived from OGTT [20]. Subjects in the upper tertile—defined as MHO (*n* = 292)—and those in the lower tertile—defined as metabolically unhealthy obese (MUHO) (*n* = 292)—were included in the study. All diabetic patients were excluded from the study. Moreover, subjects under 18 and over 65 years, pregnant or nursing, with history or clinical evidence of alcohol/drug abuse, diabetes, hypertension, coronary or peripheral artery disease, valvular heart disease, malignant disease, coagulation abnormalities, chronic gastrointestinal disease with malabsorption, chronic pancreatitis, endocrine disorders, liver or renal failure—defined as estimated glomerular filtration rate (eGFR) <60 mL/min/1.73 m^2^, on any pharmacological treatment able to affect glucose metabolism, were also excluded. All the participants underwent anamnesis, physical examination with determination of waist circumference, weight, height and BMI expressed as kg/m^2^, heart rate and measurements of systolic and diastolic blood pressure as indicated by current guidelines [21]. All the evaluations were made according to Declaration of Helsinki, after obtaining approval by local Ethical Committee and written informed consent by each subject.

### 2.2. Laboratory Determinations

Blood samples after 12-h overnight fast were obtained from all participants. Glucose, triglyceride, low (LDL) and high (HDL) density lipoprotein cholesterol concentrations were determined by enzymatic methods (Roche, Basel, Switzerland). Plasma insulin concentration was obtained with a chemiluminescence-based assay (Immulite, Siemens, Italy) and high-sensitivity C-reactive protein (hs-CRP) was measured by automated instrument (CardioPhase_hs-CRP, Siemens, Milano, Italy). Serum creatinine and uric acid (UA) were measured in the routine laboratory by an automated technique based on the measurement of Jaffe chromogen and by the URICASE/POD (Boehringer Mannheim, Mannheim, Germany) method implemented in an autoanalyzer. Values of eGFR were calculated by using the CKD-EPI equation [22].

### 2.3. Insulin-Resistance and Sensitivity

To measure insulin-resistance, the homeostasis model assessment of insulin-resistance (HOMA-IR) index was calculated as follow:(fasting insulin (μU/mL) × fasting glucose (mmol/L))/22.5(1)

Conversely, insulin sensitivity was evaluated by Matsuda index as follow:10,000/square root of (fasting glucose (mmol/L) × fasting insulin (mU/L)) × (mean glucose × mean insulin during OGTT)(2)

The Matsuda index was previously used to distinguish different metabolic obesity phenotypes [4,5], due to its strong relation to euglycemic–hyperinsulinemic clamp, which represents the gold standard test for measuring insulin sensitivity [20].

### 2.4. Echocardiographic Measurements

Tracings were taken with patients in a partial left decubitus position using a VIVID-7 Pro ultrasound machine (GE Technologies, Milwaukee, WI, USA) with an annular phased array 2.5-MHz transducer. Only frames with optimal visualization of cardiac structures were considered for readings. The mean values from at least five measurements of each parameter for each patient were computed. All the readings were performed by the same experienced investigator to optimize the reproducibility. In our laboratory, the CVs were 3.85% for posterior wall thickness, 3.70% for interventricular septum thickness, 1.50% for left ventricular internal diameter, and 5.10% for LVM. The echocardiographer was blinded to MHO/MUHO status of the subjects.

Tracings were recorded under two-dimensional guidance, and M-mode measurements were taken at the tip of the mitral valve or just below, as suggested by current guidelines [23]. The measurement of left atrial (LA) volume was performed using the area–length (L) method. We measured single-plane area of the LA from the four-chamber view, at end-ventricular systole, guaranteeing that there was no foreshortening of the LA. LA volume index (LAVI) was obtained indexing LA volume by body surface area (BSA). Left ventricular (LV) diastolic function was evaluated measuring the peak trans-valvular flow velocity in early diastole (E wave), the peak trans-valvular flow velocity in late diastole (A wave) and E-to-A ratio. Measurements of interventricular septum thickness, posterior wall thickness, and LV internal diameter (LVID) were made at end-diastole and end systole. LV end-diastolic (LVEDV) and end-systolic volume (LVESV) were measured according to Simpson method and indexed for BSA. LVM was calculated using the Devereux equation [24] and normalized by height^2,7^ [LVM index (LVMI)] according to de Simone and co-workers [25]. LV hypertrophy (LVH) was defined by a value of LVMI >44 g/m^2.7^ for females and >48 g/m^2.7^ for males [23]. Finally, to define the different left ventricular geometric patterns, the relative wall thickness (RWT) was calculated according to the following formula:(2 × posterior wall thickness)/(LVID at end-diastole)(3)

RWT allows categorization of an increase in LVM as either concentric (RWT > 0.42) or eccentric (RWT ≤ 0.42) hypertrophy and allows the identification of concentric remodeling (normal LVM with increased RWT) [23].

### 2.5. Statistical Analysis

To test the differences between MHO and MUHO groups unpaired Student’ *t*-test for clinical and biological data was performed and *χ*^2^ test was considered for categorical variables. Univariate logistic regression analysis was performed to test the effect of different covariates on LVH. MHO phenotype was considered as dichotomous variable so as gender; the other covariates were age, BMI, pulse pressure, LDL-cholesterol, uric acid, eGFR and hs-CRP. Afterwards, only variables achieving statistical significance at univariate model were included in a stepwise logistic regression model to define the independent predictors of LVH. Data are reported as mean ± standard error of the mean (SEM) and differences were considered significant at *p* < 0.05. All comparisons were performed using the statistical package SPSS 20.0 for Windows (SPSS Inc., Chicago, IL, USA).

## 3. Results

### 3.1. Study Population

Anthropometric, hemodynamic and biochemical characteristics of the whole study population and according to the different obesity phenotypes are reported in Table 1. No differences in gender distribution, age, BMI, smokers, diastolic blood pressure (DBP) and LDL-cholesterol values were observed between MUHO and MHO groups. Of interest, MUHO showed a worse hemodynamic, metabolic and inflammatory profile when compared to MHO, as suggested by significantly higher values of systolic blood pressure (SBP) and pulse pressure, heart rate, waist circumference, triglycerides, fasting glucose, fasting insulin, HOMA-IR, hs-CRP and uric acid. Conversely, MHO had lower levels of Matsuda index, HDL-cholesterol and eGFR when compared to MUHO.

### 3.2. Echocardiographic Measurements

Echocardiographic parameters of the whole study population and according to the different obesity phenotypes can be observed in Table 2. MUHO group showed significantly higher values of LAVI and left ventricular cavity size, expressed by left ventricular end-diastolic diameter (LVEDD) and LVEDV indexed for BSA (LVEDVI) (Figure 1), when compared to MHO group. Moreover, MUHO subjects had increased diastolic interventricular septum (dIVS) and reduced diastolic posterior wall (dPW), with higher values of LVMI (Figure 1). In addition, MUHO had a worse diastolic function expressed by a significantly reduced E/A ratio and a lower RWT in comparison with MHO subjects.

Patterns of left ventricular geometry of the whole study population and according to the two obesity phenotypes are showed in Table 3. The prevalence of LVH was significantly higher in MUHO than MHO (50.3% vs. 25.0%, *p* < 0.0001) (Figure 1). Among those without LVH, concentric remodeling was more common in MHO group (44.9% vs. 30.2%, *p* < 0.0001). With regards to hypertrophy pattern, the eccentric type was prevailing in MUHO group while the concentric type was more prevalent among MHO subjects (Figure 2), suggesting a different ventricular adaptation in relation to obesity phenotype.

### 3.3. Logistic Regression Analysis on LVH Risk

A logistic regression analysis was performed to test the effect of different variables, including MHO, on the risk of LVH (Table 4). At univariate model, both gender and age were associated to higher risk of LVH, respectively doubled for males and increased of 25% for every 10 years of aging. Conversely, MHO phenotype and eGFR resulted to be protective factors on LVH development; in particular, probability of LVH decreases by 57% in MHO subjects and 9% for every 10 mL/min/m^2^ of increase in eGFR. These results were mostly confirmed at multivariate model, with protective effect of MHO (OR 0.46; *p* < 0.0001) and eGFR (OR 0.91; *p* = 0.009), and increased risk for age (OR 1.22; *p* = 0.001) and male gender (OR 2.64; *p* < 0.0001).

## 4. Discussion

In this study, we have demonstrated a strong association between obesity and LVH in a large cohort of obese subjects without history of cardio-metabolic disease. The prevalence of LVH was 38.5% in our study population. Most importantly, the association between obesity and increased cardiac mass was strongly affected by metabolic phenotype, but not by BMI. Indeed, our data showed that LVH was more prevalent in MUHO than MHO subjects (50.3% vs. 25.0%) and the risk for LVH decreased by 57% in MHO group. Another interesting finding is the different ventricular remodeling pattern among the groups: MHO group had a more prevalent concentric pattern in comparison with MUHO group, which instead showed a higher prevalence of eccentric geometry. Of interest, MUHO had also a worse diastolic function, expressed by lower E/A ratio, when compared to MHO.

On pathophysiological side, relationship between adipose tissue and cardiac damage appears extremely complex and our data contribute to clarify the underlying mechanisms. As expected, MUHO group had a less favorable cardio-metabolic profile due to higher insulin-resistance. In our study, MUHO subjects—compared to MHO matched for age, gender and BMI—had significantly higher values of waist circumference, triglycerides, SBP, pulse pressure and hs-CRP, as well as lower levels of HDL-cholesterol. Taken together, these anthropometric and biochemical data are expression of insulin-resistance as well as the most important features of metabolic syndrome [26,27]. It has been demonstrated that central obesity, clinically expressed as elevated waist circumference, is characterized by reduced insulin sensitivity that in turn leads to free fatty acid flux to the liver with increased release of lipoprotein rich in triglycerides [28]; moreover, abnormal activity of adipocyte in central obesity leads to enhanced subclinical inflammation that is strongly associated with insulin resistance [29,30]. Finally, accumulating evidence showed the strong association between insulin-resistance and the continuum of CV disease, from endothelial dysfunction [31,32,33,34,35] to vascular [36,37] and cardiac damage [38], further corroborating the importance of metabolic profile in different settings of patients.

In view of this we have demonstrated that, in obese individuals, LVH and its geometry do not necessarily change directly with the expansion of the adipose tissue but may be affected by different metabolic phenotype, with important implications on clinical outcome and therapeutic strategy. We can suppose that the prevalent eccentric pattern of LVH in MUHO individuals may be attributable to the predominance of the obesity-related volume overload that determines elongation of cardiomyocytes by sarcomeric addition in series as adaptive mechanism and, consequently, an eccentric LVH. This volume overload is probably related to an excessive hydrosaline retention determined by insulin-resistance and reduced renal function, as evidenced by significantly lower levels of eGFR in MUHO group. Logistic regression analysis confirmed the role of renal function on left ventricular geometry: for each increase of 10 mL/min/m^2^ of eGFR, LVH risk decreased by 9%. Regarding insulin-resistance, the associated hyperinsulinemia directly increases cardiac mass and left ventricle dysfunction, through the interaction between insulin, its receptor and insulin-like growth factor-1 (IGF-1) receptor, expressed in the myocardium [39]. Moreover, insulin-resistance reduces IGF-1 blood levels, for negative balance and spill-over effect between hormones. Reduced levels of IGF-1 are associated with higher circulating levels of growth hormone and insulin that promote hypertrophy and myocardial fibrosis [8,40]. Moreover, lower amounts of plasma IGF-1 could contribute to the reduced eGFR observed in MUHO individuals as compared with MHO ones. Previous studies have in fact reported that IGF-1 increases both renal blood flow and renal filtration by stimulating IGF-1 receptors [41,42]. IGF-1 is a potent vasodilator by stimulating nitric oxide biosynthesis through up-regulation of endothelial nitric oxide synthase expression in endothelial cells. According to these pathophysiological mechanisms, recent evidence showed that MUHO individuals have a 2.5 fold increased risk of impaired renal function compared to MHO subjects, and a 7.0 fold increased risk compared with non-obese subjects [5]. Insulin-resistance also determines the hyperactivation of other systems such as renin-angiotensin-aldosterone system (RAAS) and symphatetic nervous system, with consequent hemodynamic and no hemodynamic effects. Furthermore, our group has demonstrated that there is a strong association between increased RAAS attivation, circulating insulin levels and LVH [14]. This is promoted by the harmful effect of angiotensin II, which can increase the expression of profibrotic substances, such as plasminogen activator inhibitor type-1 and transforming growth factor-β, in addition to the direct profibrotic effect of aldosterone. To confirm this, the use of the angiotensin receptor blocker losartan in obese rats reduces the risk of myocardial fibrosis and LVH. This may be determined by the improvement of angiotensin II-related insulin-resistance, as evidenced by reduction of insulin levels and by improvement of glucose parameters [43]. Insulin-resistance induced by angiotensin II, in turn, promotes further activation of RAAS and symphatetic nervous system, feeding a vicious circle that promotes myocardial damage [33]. Moreover, such authors have documented that insulin-resistance induces the shift of cardiac metabolism towards free fatty acid oxidation, leading to lower myocardial energy efficiency, mitochondrial dysfunction, reactive oxygen species production and cardiomyocytes senescence, apoptosis and fibrotic substitution [44]. Finally, all these mechanisms can further favour a condition of mild chronic inflammation, that in MUHO may be facilitated by an increased visceral adipose tissue, with systemic and also paracrine secretion of inflammatory adipokines such as tumor necrosis factor alpha, monocyte chemoattractant protein-1 and interleukin-6, that may further increase cardiac damage and remodeling [45]. The present study showed that the interaction between all these systems is not the same among obese subjects, and depends by different metabolic phenotypes. Therefore, cardiac remodeling in the obese is very complex and involves different aspects. Obesity is an independent risk factor for LVH, but it is also associated with other risk factors, such as hypertension and sleep apnea, which can amplify the effect of obesity on cardiac remodeling, through a combination of volume and pressure overload. This may have consequences on LVH and its geometry pattern, influencing left ventricular cavity and parietal thicknesses [46].

The results of the present study have great biological plausibility because, although the structural remodeling of the myocardium represents an adaptive factor able to guarantee good functional performances in various systemic pathologies, including obesity, it is also true that LVH is recognized as an independent predictor of CV events. Among the organ damage indicators, LVH is an independent risk factor for mortality and morbidity and is the only indicator of organ damage whose regression is associated with improved prognosis. It is known that, in patients with LVH, the reduction of LVM is associated with favorable pathophysiological changes such as improvement of systolic function parameters and diastolic filling, increase of coronary reserve, reduction of ventricular arrhythmias and, probably, prevention of atrial fibrillation [47].

Our data support the hypothesis that the different metabolic phenotypes of obesity have a different effect on the left ventricular geometry, which is strongly influenced by the metabolic setting. This may have important therapeutic implications suggesting the usefulness, in MUHO patients, of drugs able to reduce the volume overload, such as diuretics, as well as sodium restriction. Furthermore, it is recommended to use, in obese patients, drugs that can modulate RAAS, such as ACE inhibitors or angiotensin receptor blockers, that are able to reduce blood pressure and cardiac remodeling. In addition, weight loss is fundamental. In fact, a recent case-control study [48] showed that the reduction of body weight determines a positive cardiac remodeling, as evidenced by the improvement of ventricular volumes, wall thickness and ventricular mass. We do not know the prognostic impact of different ventricular remodeling pattern observed in MHO individuals, compared with MUHO. Moreover, this pattern may evolve, because of a metabolic profile worsening, from a concentric remodeling towards a dilated type and heart failure. A better phenotypic characterization of obese subjects can be useful in order to evaluate their cardio-metabolic risk profile and to use more targeted therapeutic approaches.

## 5. Conclusions

Our study demonstrates that obese subjects—matched for age, gender and BMI—have different LVM and geometry due to different insulin sensitivity status. MHO develop more commonly a concentric remodeling, and have a reduced risk for LVH when compared to MUHO, in which the eccentric type is more prevalent. Therefore, these results contribute to better understand the pathophysiological mechanisms underlying the cardiac damage in obese subjects, suggesting that different metabolic phenotypes are associated with different myocardial adaptation.

### Strengths and Limitations

A strong point of the present study is represented by having excluded diabetic and hypertensive pharmacological treatment that could interfere with cardiac remodeling. Furthermore, metabolic profile was evaluated through the insulin sensitivity index Matsuda, strongly correlated to euglycemic clamp and LVM was indexed for height^2,7^ that is a more accurate method to avoid underestimating LVH in obese subjects. However, this is a cross-sectional study therefore no causal link can be demonstrated. Another possible limitation of the study may be the absence of information on the presence of sleep apnea, which can further amplify the effect of obesity on cardiac remodeling. Finally, prospective studies on larger populations are needed to evaluate how ventricular remodeling evolves over time in MHO subjects and its impact on CV risk.

## Figures and Tables

**Figure 1 nutrients-12-00412-f001:**
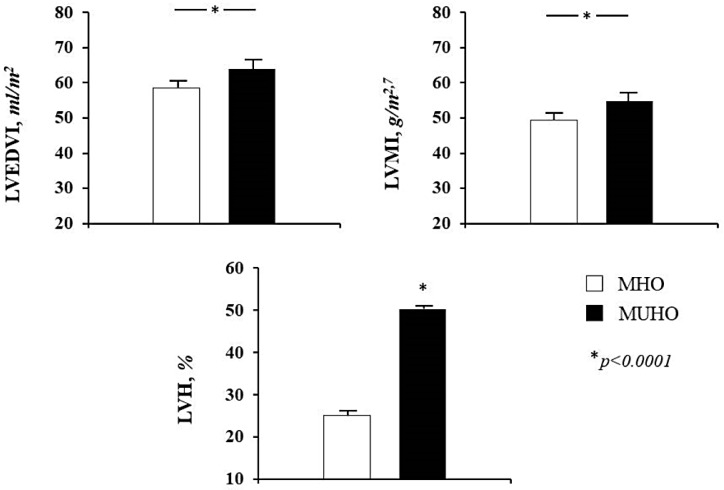
Mean values of left ventricular end-diastolic volume indexed for body surface area (LVEDVI), left ventricular mass index (LVMI) and percentage of patients with left ventricular hypertrophy (LVH) according to obesity phenotype. MHO = metabolically healthy obese; MUHO = metabolically unhealthy obese.

**Figure 2 nutrients-12-00412-f002:**
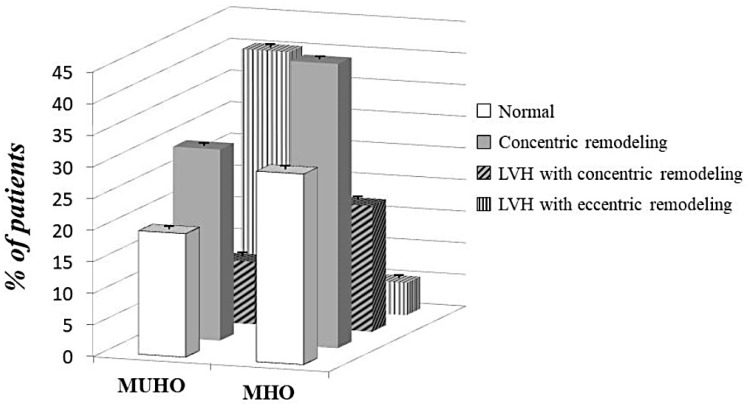
Different patterns of left ventricular geometry according to obesity phenotypes. MHO = metabolically healthy obese; MUHO = metabolically unhealthy obese; LVH = left ventricular hypertrophy. Differences between groups: *p* = 0.003 for normal pattern; *p* = 0.001 for LVH with concentric pattern; *p* < 0.0001 for other patterns.

**Table 1 nutrients-12-00412-t001:** Anthropometric, hemodynamic and biochemical characteristics of the whole study population and according to different obesity phenotypes.

	All (*n* = 876)	MHO (*n* = 292)	MUHO (*n* = 292)	*p*
Gender m/f	380/496	127/165	132/160	0.677 *
Age, years	48.3 ± 0.5	47.4 ± 0.8	48.3 ± 0.9	0.454
Smokers, *n* (%)	171 (19.5)	61 (20.9)	57 (19.5)	0.680 *
BMI, kg/m^2^	36.0 ± 0.2	35.0 ± 0.4	35.9 ± 0.3	0.080
WC, cm	109.6 ± 0.4	106.7 ± 0.6	113.4 ± 0.9	<0.0001
SBP, mmHg	132.7 ± 0.5	130.5 ± 0.7	134.3 ± 1.1	0.005
DBP, mmHg	85.1 ± 0.3	84.9 ± 0.5	85.5 ± 0.6	0.478
Pulse Pressure, mmHg	47.6 ± 0.5	45.6 ± 0.7	48.9 ± 1.1	0.012
Heart rate, bpm	72.8 ± 0.4	69.8 ± 0.6	75.7 ± 0.7	<0.0001
LDL-cholesterol, mg/dL	124.6 ± 1.2	124.4 ± 2.1	124.8 ± 2.3	0.913
HDL-cholesterol, mg/dL	49.2 ± 0.4	51.1 ± 0.7	45.5 ± 0.7	<0.0001
Triglycerides, mg/dL	128.9 ± 2.2	116.6 ± 3.4	149.5 ± 4.6	<0.0001
Serum creatinine, mg/dL	0.8 ± 0.01	0.7 ± 0.01	0.8 ± 0.01	0.002
eGFR, mL/min/1.73 m^2^	108.5 ± 0.9	118.8 ± 1.9	96.5 ± 1.1	<0.0001
Uric acid, mg/dl	5.2 ± 0.05	4.9 ± 0.08	5.6 ± 0.09	<0.0001
Fasting glucose, mg/dl	94.6 ± 0.5	90.9 ± 0.6	99.6 ± 0.9	<0.0001
Fasting insulin, µU/ml	11.6 ± 0.2	9.1 ± 0.2	15.6 ± 0.3	<0.0001
Matsuda index	62.6 ± 0.9	91.4 ± 1.9	39.1 ± 0.5	<0.0001
HOMA-IR	3.6 ± 0.1	2.1 ± 0.1	6.4 ± 0.2	<0.0001
hs-CRP, mg/L	3.5 ± 0.1	3.0 ± 0.1	3.9 ± 0.2	<0.0001

* by *χ*^2^ test. MHO = metabolically healthy obese; MUHO = metabolically unhealthy obese; BMI = body mass index; WC = waist circumference; SBP = systolic blood pressure; DBP = diastolic blood pressure; LDL = low-density lipoprotein; HDL = high-density lipoprotein; eGFR = estimated glomerular filtration rate; HOMA-IR = homeostatic model assessment of insulin resistance; hs-CRP = high-sensitivity C-reactive protein.

**Table 2 nutrients-12-00412-t002:** Echocardiographic parameters of the whole study population and according to different obesity phenotypes.

	All (*n* = 876)	MHO (*n* = 292)	MUHO (*n* = 292)	*p*
LAVI, mL/m^2^	29.7 ± 0.4	26.9 ± 0.4	32.1 ± 0.7	<0.0001
LVEDD, cm	4.92 ± 0.01	4.83 ± 0.02	5.01 ± 0.02	<0.0001
LVEDVI, mL/m^2^	61.7 ± 0.5	58.5 ± 0.7	63.9 ± 0.9	<0.0001
dPW, cm	1.01 ± 0.01	1.03 ± 0.01	0.99 ± 0.01	0.029
dIVS, cm	1.12 ± 0.01	1.10 ± 0.01	1.15 ± 0.01	<0.0001
LVMI, g/m^2.7^	52.5 ± 0.6	49.3 ± 1.1	54.9 ± 1.2	<0.0001
E/A	0.96 ± 0.01	1.04 ± 0.02	0.87 ± 0.02	<0.0001
RWT	0.41 ± 0.01	0.43 ± 0.01	0.40 ± 0.01	<0.0001

MHO = metabolically healthy obese; MUHO = metabolically unhealthy obese; LAVI = left atrial volume index; LVEDD = left ventricular end-diastolic diameter; LVEDVI = left ventricular end-diastolic volume index; dPW = diastolic posterior wall; dIVS = diastolic interventricular septum; LVMI = left ventricular mass index; RWT= relative wall thickness.

**Table 3 nutrients-12-00412-t003:** Patterns of left ventricular geometry in the whole study population and according to different obesity phenotypes.

	All (*n* = 876)	MHO (*n* = 292)	MUHO (*n* = 292)	*p*
LVH–, *n* (%)	539 (61.5)	219 (75.0)	145 (49.7)	<0.0001
Normal, *n* (%)	212 (24.2)	88 (30.1)	57 (19.5)	0.003
Concentric remodeling, *n* (%)	327 (37.3)	131 (44.9)	88 (30.2)	<0.0001
LVH+, *n* (%)	337(38.5)	73 (25.0)	147 (50.3)	<0.0001
Eccentric remodeling, *n* (%)	206 (23.5)	15 (5.1)	118 (40.4)	<0.0001
Concentric remodeling, *n* (%)	131 (15.0)	58 (19.9)	29 (9.9)	0.001

MHO = metabolically healthy obese; MUHO = metabolically unhealthy obese; LVH = left ventricular hypertrophy.

**Table 4 nutrients-12-00412-t004:** Univariate and multivariate stepwise logistic regression analysis with left ventricular hypertrophy as dependent variable, in whole study population.

	Univariate		Multivariate	
	Odds Ratio	*p*	Odds Ratio	*p*
MHO phenotype, yes/no	0.43	<0.0001	0.46	<0.0001
eGFR, 10 mL/min/1.73 m^2^	0.91	0.008	0.91	0.009
Age, 10 years	1.25	<0.0001	1.25	<0.0001
Gender, male	2.62	<0.0001	2.64	<0.0001
BMI, kg/m^2^	1.02	0.131		
Pulse pressure, 10 mmHg	1.03	0.631		
LDL-cholesterol, 10 mg/dL	0.99	0.844		
Uric acid, mg/dL	0.96	0.556		
hs-CRP, mg/L	0.95	0.159		

MHO = metabolically healthy obese; eGFR = estimated glomerular filtration rate; BMI = body mass index; LDL = low-density lipoprotein; hs-CRP = high-sensitivity C-reactive protein.

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
