# Peer review of "Different Patterns of Left Ventricular Hypertrophy in Metabolically Healthy and Insulin-Resistant Obese Subjects"

_nutrients, 2020, doi:10.3390/nu12020412_

Round 1
Reviewer 1 Report
The manuscript by Sciacqua et al., describes the LV remodeling in a large cohort of metabolically healthy obese subjects (MHO) and metabolically unhealthy obese adults (MUHO). The authors found that there was a higher prevalence of LVH in the latter - and a higher prevalence of eccentric hypertrophy in this group. The MHO group had, by contrast, a higher prevalence of concentric LV remodeling. Stepwise multivariable modeling showed that phenotype (MHO/MUHO) had an independent effect of the risk of LVH.
The strengths of this study are 1) its large size, and 2) the categorization of patients as MHO or MUHO based on Matsuda index.
Specific questions:
1) Diabetes should be explicitly listed as an exclusion critieria in the Methods.
2) Were the echocardiographers blinded to the MHO/MUHO status of the subjects? If so, this should be stated.
3) Table 4. should state that the multivariate regression was stepwise in the Table title.
3a) line 236 ? the prevalence of LVH was 38.5% in our study population, according to previous evidence? The LVH prevalence in this study was measured - and should not be according to a previous study.
4) line 262 'can be attributable to the predominance of the obesity-related volume overload..." I don't think this has been proven in this manuscript, but is rather a supposition. The wording of this sentence should reflect that this is a supposition.
5) line 271 ...expressed in the myocardium" needs a reference.
Line 279 and 280 I think the authors mean 2.5 fold and 7 fold
Author Response
Point 1: Diabetes should be explicitly listed as an exclusion critieria in the Methods.
Response 1: We have clearly reported in the materials and methods section, that diabetes is an exclusion criteria (line 78).
Point 2: Were the echocardiographers blinded to the MHO/MUHO status of the subjects? If so, this should be stated.
Response 2: The echocardiographer was blinded to MHO/MUHO status of the patients. We have reported this statement in the materials and methods section (lines 119,120).
Point 3: Table 4. should state that the multivariate regression was stepwise in the Table title.
Response 3: We have corrected the table 4 title as requested.
Point 3a: line 236 ? the prevalence of LVH was 38.5% in our study population, according to previous evidence? The LVH prevalence in this study was measured - and should not be according to a previous study.
Response 3a: We have removed this sentence, as suggested (line 238).
Point 4: line 262 'can be attributable to the predominance of the obesity-related volume overload..." I don't think this has been proven in this manuscript, but is rather a supposition. The wording of this sentence should reflect that this is a supposition.
Response 4: We have reworded the sentence, as suggested (line 264).
Point 5: line 271 ...expressed in the myocardium" needs a reference.
Response 5: We have added the reference, as requested (reference 39).
Point 6: Line 279 and 280 I think the authors mean 2.5 fold and 7 fold
Response 6: We have corrected the sentence, as suggested (lines 282, 283).
Reviewer 2 Report
In Methods, please add the reference to “LV hypertrophy (LVH) was defined by a value of LVMI >44 g/m2.7 for females and >48 g/m2.7 for males.”. It is better to present the data as mean±SEM. In both Figure 1 and Figure 2, there are two bar graphs without error bar. Please do correction. For logistic regression analysis on LVH risk, two different statistical analysis methods were used. Univariate and Multivariate cannot be used together. Pease explain it. In Discussion, it is concluded that MHO group 240 had a more prevalent concentric pattern in comparison with MUHO group. However, in Introduction, it is possible to speculate that it is not obesity itself, but the obesity-related metabolic abnormalities that affect LVM and LVH development with possible consequence on clinical outcome. It will make the reader confused. Inflammation plays important roles in different pathophysiologic process. MUHO showed a worse inflammatory profile when compared to MHO. Please add more discussion about the role of inflammation in different pattern of left ventricular hypertrophy in metabolically healthy and insulin-resistant obese subjects. There are a number of typographical errors and misspellings.
Author Response
Point 1: In Methods, please add the reference to “LV hypertrophy (LVH) was defined by a value of LVMI >44 g/m2.7 for females and >48 g/m2.7 for males.”
Response 1: We have added the reference, as suggested (reference 23).
Point 2: It is better to present the data as mean±SEM.
Response 2: We have calculated SEM for all mean values and we specified this issue in the statistical analysis section (line148). Moreover, according with this, we have revised the figure presenting the error bars.
Point 3: In both Figure 1 and Figure 2, there are two bar graphs without error bar. Please do correction.
Response 3: I made the corrections both in figure 1 and figure 2, as suggested, the new version of the figures has been uploaded with the revision.
Point 4: For logistic regression analysis on LVH risk, two different statistical analysis methods were used. Univariate and Multivariate cannot be used together. Pease explain it.
Response 4: Thanks for the suggestion. Univariate logistic regression analysis was performed to test the effect of different covariates on LVH. In a second step, only variables achieving statistical significance at univariate model were included in a stepwise logistic regression model to define the independent predictors of LVH. We clarified this issue in the materials and methods, statistical analysis, section (lines 143-148).
Point 5: In Discussion, it is concluded that MHO group 240 had a more prevalent concentric pattern in comparison with MUHO group. However, in Introduction, it is possible to speculate that it is not obesity itself, but the obesity-related metabolic abnormalities that affect LVM and LVH development with possible consequence on clinical outcome. It will make the reader confused. Inflammation plays important roles in different pathophysiologic process. MUHO showed a worse inflammatory profile when compared to MHO.Please add more discussion about the role of inflammation in different pattern of left ventricular hypertrophy in metabolically healthy and insulin-resistant obese subjects.
Response 5: We revised the discussion as suggested and we edited the references accordingly (lines 298-302, reference 45).
Point 6: There are a number of typographical errors and misspellings.
Response 6: We have revised the whole manuscript for correcting the typographical errors and misspellings.